# Sustained Release of Azoxystrobin from Clay Carriers for the Management of Maize Late Wilt Disease

**DOI:** 10.3390/jof12010021

**Published:** 2025-12-27

**Authors:** Ofir Degani, Adar Abramovici, Achinoam Levi-Lion, Daniel Demenchuk, Ariel Hadad, Elhanan Dimant

**Affiliations:** 1Faculty of Sciences, Tel-Hai College, Upper Galilee, Tel-Hai 1220800, Israel; adarabra@gmail.com (A.A.); achinoaml56@gmail.com (A.L.-L.); trinitywolf27@hotmail.com (D.D.); arielhadad1996@gmail.com (A.H.); 2Migal–Galilee Research Institute, Tarshish 2, Kiryat Shmona 1101600, Israel; elhanand@migal.org.il

**Keywords:** bentonite, *Cephalosporium maydis*, disease control, bioassay, *Harpophora maydis*, *Magnaporthiopsis maydis*, sepiolite, slow-release

## Abstract

Controlled-release technologies based on natural clays offer a sustainable approach to enhance the efficacy and environmental compatibility of agrochemicals. This study reports the development and evaluation of clay-based azoxystrobin (Az) formulations for controlling *Magnaporthiopsis maydis*, the causal agent of maize late wilt disease. Among six carriers tested, raw bentonite and sepiolite were selected for their comparable adsorption capacity (9.5% Az loading efficiency) and ease of preparation. A novel mycelial plug-immersion bioassay was established and calibrated (R^2^ = 0.92–0.95) to assess release kinetics and antifungal efficacy, showing approximately tenfold higher sensitivity than conventional disk-diffusion or mycelial-growth inhibition assays. Sequential wash and extended incubation experiments demonstrated sustained Az release equivalent to ≥1 mg L^−1^ over 144 h, resulting in approximately 50% (*p* < 0.05) fungal growth suppression. A comparative analysis of particle suspensions and supernatants revealed formulation-specific release behaviors, which differed among clay carriers. Overall, bentonite and sepiolite acted as efficient carriers that prolonged fungicide bioavailability, minimized leaching losses, and preserved biological activity. These findings provide proof of concept for clay–Az formulations as eco-friendly and cost-effective tools for late wilt management and advance understanding of clay–fungicide interactions that support sustainable, integrated disease-control strategies.

## 1. Introduction

Maize (*Zea mays* L.) is the leading global grain crop, alongside rice and wheat [1,2]. Maize serves as a staple food for humans and a primary fodder source for animals, while also supporting diverse industries through the production of food additives, oils, starch, paper, and biofuels [1]. However, the full yield potential of maize is often constrained by diseases [3]. Among these, late wilt disease (LWD), caused by the fungus *Magnaporthiopsis maydis* (formerly *Harpophora maydis* and *Cephalosporium maydis*), is a severe threat to commercial maize production in highly impacted areas such as Egypt, Israel, India, Portugal, and Spain [4,5].

The pathogen penetrates maize plants at the onset of the growing season, colonizes the vascular tissues, and obstructs water transport to the upper parts of the plant, leading to plant collapse approximately two weeks before harvest [6]. In some fields and in susceptible cultivars, disease incidence may reach 50–100% [7,8,9]. Drought stress is recognized as a critical factor that further compromises plant resistance to diseases [10]. The persistence of the fungus in soil [11], maize residues [12,13], seeds [14,15], and alternative host plants [16,17,18] poses a major barrier to effective disease management.

The predominant LWD management relies on agrotechnical measures that integrate avoidance, exclusion, and eradication principles [19]. Despite the promising potential of these approaches, the only method currently implemented in highly impacted areas to control the disease is the use of resistant maize cultivars [20]. Over the past few decades, breeding programs have been conducted in Israel [21], Egypt [22,23], and India [24] to identify and develop resistant LWD varieties. However, aggressive pathogen strains capable of breaking the resistance of these lines have been reported in Egypt, Spain, and Israel [25,26,27,28].

A promising approach for managing LWD in severely affected areas involves the controlled application of azoxystrobin (Az) through drip irrigation at three intervals (15, 30, and 45 days after sowing) [6,29]. Adapting the cultivation method to paired rows facilitates efficient movement of the fungicide from the dripline to the plant roots, while substantially reducing application costs. In heavily infested fields with susceptible cultivars, this treatment eliminated pathogen DNA from root and stalk tissues, delayed symptom onset by 41%, and improved overall crop yield and quality by 36% and 77%, respectively—restoring them to levels observed in healthy fields. Furthermore, combining Az with preparations of different modes of action proved effective and may reduce the risk of resistance development [30]. However, modifications in cultivation practices and reliance on drip irrigation are not universally applicable. Moreover, the use of chemical pesticides raises concerns due to their potential environmental and health risks [31]. In addition, the pathogen may eventually develop resistance to Az, further limiting the long-term effectiveness of this approach [32,33].

Consequently, considerable efforts are currently directed toward developing environmentally friendly strategies for managing maize late wilt [30,34,35,36]. While green pesticides provide important advantages, their efficacy is often constrained under severe disease pressure and remains highly dependent on favorable environmental conditions [37]. An integrated approach, combining biological agents (e.g., *Trichoderma* spp. and beneficial rhizobacteria) with low doses of chemical fungicides (e.g., Az), has shown promise by enhancing the stability and effectiveness of bio-agents in fluctuating and unfavorable environments [6,30].

Clay minerals have been widely reported to act as platforms for controlled pesticide delivery due to their diverse morphologies and sorption capacities [38,39,40,41]. Collectively, these studies support the view that clay-based carriers can prolong fungicide bioavailability, mitigate environmental contamination, and enhance disease management efficacy under real conditions. As a result, clay-based delivery systems emerge as promising and sustainable alternatives to conventional pesticide application, in line with the growing trend toward precision agriculture and reduced chemical loading. A recent example is the successful application of clay-based carriers for the controlled release of essential oils in net house-grown chives, achieving thrips management without phytotoxic effects [42]. In the present study, we did not directly characterize clay–fungicide interactions; instead, we evaluated their functional consequences through biological release assays.

The advantages of clay-based carriers for pesticide delivery in managing LWD remain largely unexplored. Clays represent an optimal platform for this purpose, as they are naturally abundant, inexpensive, and generally regarded as environmentally benign [43]. These carriers can be employed in their raw form or functionalized with organic cations to enhance adsorption capacity, release kinetics, and bioactivity. Accordingly, clay-based formulations—applied either as seed coatings or as strip placements at sowing—offer a cost-effective alternative to conventional chemical treatments [44]. Such approaches can be readily implemented under diverse field conditions without requiring specialized infrastructure. The gradual release of Az from clay particles positioned in the sowing strip can provide extended protection of maize seedlings during their critical window of vulnerability to fungal establishment, while simultaneously minimizing fungicide losses via leaching or lateral soil movement. Moreover, this localized delivery system can facilitate the integration of chemical and biological control strategies, thereby further reducing pesticide use. Importantly, clay–Az formulations may also extend their protective spectrum to other soil-borne pathogens, thereby broadening their agronomic relevance.

The current study introduces an innovative approach that integrates materials science and plant pathology to develop a sustainable, clay-based controlled-release system for Az delivery against *Magnaporthiopsis maydis*, the causal agent of maize LWD. To this end, sepiolite and bentonite were selected and systematically examined for their adsorption capacity, sustained-release kinetics, and antifungal efficacy against *M. maydis*. The work pioneers the use of a novel mycelial plug-immersion bioassay, enabling precise quantification of slow-release dynamics and fungicidal activity. Unlike conventional fungicide applications that rely on drip irrigation or repeated treatments, the proposed clay–Az formulations—based on natural, inexpensive, and ecologically compatible minerals—allow gradual fungicide release directly in the root zone, maintaining prolonged antifungal activity while minimizing chemical input and leaching losses. Collectively, these advances aimed to establish a proof of concept for clay-mediated, controlled-release fungicidal systems, opening new perspectives for integrating chemical and biological control strategies toward sustainable management of soil-borne crop diseases.

## 2. Materials and Methods

### 2.1. Rationale and Research Design

The experimental design followed a hypothesis-driven, stepwise workflow in which each assay addressed a specific question and informed the next stage. Conventional mycelial-growth inhibition and disk-diffusion assays were first used to (i) characterize azoxystrobin (Az) sensitivity among *Magnaporthiopsis maydis* isolates and (ii) evaluate the limitations of standard plate-based methods at low fungicide concentrations. These assays served as screening and benchmarking tools but proved insufficient to resolve low-dose or gradual fungicide release. Clay–Az formulations were then screened to select suitable carrier materials with consistent behavior and strong antifungal activity. Based on these findings, a mycelial plug-immersion bioassay was developed and calibrated to test the central hypothesis of the study: that clay carriers enable sustained, biologically effective release of Az at sub-milligram concentrations. This assay was subsequently applied to quantify release kinetics from selected clay carriers using sequential-wash experiments, providing a sensitive and mechanistically relevant assessment of controlled-release performance.

### 2.2. Fungi Used in This Study

All *M. maydis* isolates were recovered from symptomatic maize plants sampled in commercial fields in Israel. The isolates have been previously characterized by pathogenic and physiological traits, colony morphology, micromorphology, and molecular identity [25,45,46]. Isolate aggressiveness was quantified earlier in full-season, net-house potted-plant assays [25,28]. A moderately aggressive isolate, Mm2 (CBS 133165, deposited in the CBS-KNAW Fungal Biodiversity Center, Utrecht, The Netherlands), and a highly virulent isolate (Mm30) were selected for the plug-immersion bioassay study.

### 2.3. Growth of the Fungi

*M. maydis* colonies were grown on a PDA medium (Potato Dextrose Agar, Difco, Detroit, MI, USA) in 90 mm plates. The plates were kept in an incubator at a temperature of 28 ± 1 °C in the dark for 3–6 days. To transfer the fungal colony to a fresh growth plate, a 6 mm diameter disk, obtained from the perimeter of a mature colony, was removed and placed in the center of a new PDA medium and incubated at the above conditions. For submerged cultures, ten fungal disks were sown in an Erlenmeyer flask containing 150 mL of potato dextrose broth (PDB; Difco Laboratories, Detroit, MI, USA). The flasks were plugged with a breathable cover and incubated for six days, shaken in the dark at 150 rpm at 28 ± 1 °C.

### 2.4. Azoxystrobin Growth Inhibition Assay for M. maydis Isolates

The antifungal activity of Az against *M. maydis* isolates was evaluated using a mycelial disk growth inhibition assay, as described before [30]. Briefly, 90 mm Petri dishes containing PDA supplemented with Amistar SC (25% *w*/*v*, Az, Syngenta, Basel, Switzerland; supplied by Adama Makhteshim, Ashdod, Israel) containing Az at 0.008 mg L^−1^ were prepared, alongside a control medium without the fungicide. From the actively growing margins of 4–6-day-old colonies, 6 mm diameter mycelial disks of each of the 16 *M. maydis* isolates (Mm2, Mm6, Mm7, Mm14, Mm16, Mm18, Mm20, Mm24, Mm25, Mm26, Mm27, Mm28, Mm29, Mm30, Mm32, and Mm34) were excised and aseptically placed in the center of the plates. Each treatment included three replicates. Plates were incubated in the dark at 28 ± 1 °C for three days, after which the colony diameter was measured in two perpendicular directions, and the mean values were recorded. The growth of the colonies on fungicide-amended media was measured to determine the inhibitory effect of Az. The antifungal activity was quantified as the percentage inhibition of radial growth relative to the untreated control, calculated using the formula: % inhibition = [(*C* − *T*)/*C*] × 100, where *C* and *T* represent the mean colony diameters on control and treated plates, respectively. The resulting inhibition values were then used to assess and compare the relative sensitivity of the different *M. maydis* isolates to Az.

### 2.5. Disk-Diffusion Assay

Antifungal agar disk-diffusion assays were performed following the CLSI disk-diffusion guidance for fungi [47]. An *M. maydis* inoculum was prepared by growing colonies on PDA in three 90 mm Petri dishes in the dark at 28 ± 1 °C for 6 days; 1 mL of sterile double-distilled water (DDW) was added per dish, and the mycelial surface was gently scraped with a sterile Drigalski spatula to release propagules. The pooled suspension was transferred sequentially between three plates to maximize colony-forming unit (CFU) yield. A fresh PDA Petri plates (90 mm) were uniformly seeded by spreading 20 µL of a conidial–mycelial suspension (1.77–1.99 × 10^6^ spores mL^−1^ with mycelial fragments) across the surface. Commercial Amistar SC (25% *w*/*v* Az) was diluted in sterile DDW to obtain the following Az concentrations: 0.5, 0.8, 1, 1.2, 2, 3, 4, 5, and 10 mg L^−1^. A sterile 6 mm paper disk (Whatman No. 1, Cytiva, Maidstone, UK) impregnated with Az was positioned at the plate center; Water-only disks served as negative controls. Plates were incubated in the dark at 28 ± 1 °C for 6 days. Antifungal activity was quantified as the inhibition zone diameter (mm) around the disk, with three biological replicates per treatment.

#### 2.5.1. Preparation of the Clay–Az Formulation

The formulation was prepared at room temperature (~25 °C) by mixing 300 mg of sterilized or non-sterilized bentonite or sepiolite (Table 1) with 30 mL of sterile DDW (solid–liquid ratio = 10 g L^−1^). To this, 240 µL of commercial Amistar SC (25% *w*/*v* Az) was added, resulting in an Az-to-clay ratio of 1:5 (*w*/*w*), corresponding to 20% (*w*/*w*) Az relative to the clay mass. The mixture was stirred with a sterilized spatula until a smooth, homogeneous slurry formed. The slurry was spread as a thin layer on baking paper (to maximize surface area) and left under the biosafety hood, overnight (~36 h) until completely dry (to constant mass). The dried material was then ground and sieved using a sterile mortar and pestle to obtain a fine powder.

The nominal loading capacity (LC, mg Az g^−1^ clay) and encapsulation efficiency (EE, %) were determined by weight-based calculations of Amistar SC added to known clay masses, yielding LC ≈ 446–447 mg g^−1^ for bentonite and sepiolite (EE ≈ 100%) and 46 mg g^−1^ for kaolinite. Sustained release and retention were independently supported by sequential-wash bioassays in which antifungal activity persisted in both particle suspensions and clarified supernatants over 144 h.

#### 2.5.2. Inhibition of *M. maydis* Spore Germination and Hyphal Growth by Clay–Az Formulations

A modified version of the disk-diffusion assay was performed using clay–Az powder in place of an Az-soaked paper disk. The inhibitory activity of clay–Az preparations against *M. maydis* was assessed on PDA plates under different sterilization and antibiotic conditions to minimize contamination risk while using the formulation as-is. Treatments included formulations with or without Amistar SC, with or without Hygromycin B (50 µg mL^−1^, Mercury, Rosh HaAyin, Israel) incorporated into the PDA medium, and controls without clay or with Az-free clay. Plates were prepared and incubated as described above (Section 2.5). Spore concentration (1.81 × 10^6^ spores mL^−1^) was determined using a hemocytometer. Clay preparations were placed at the center of each plate with a sterile spatula, and plates were incubated at 28 ± 1 °C in darkness for six days. Inhibition zones were then measured as the diameter of fungal growth suppression surrounding the clay deposit.

### 2.6. Mycelial-Growth Inhibition Assay for Clay–Az Formulations

A mycelial-growth inhibition assay [45,47,48] was adjusted to evaluate clay–Az formulations against the late-wilt pathogen *M. maydis*. Clay–Az granules were prepared so that 50 mg of clay contained 0.4 mL of commercial Amistar SC (25% *w*/*v* Az), i.e., 100 mg active ingredient (a 1:2 pesticide: clay ratio). The examined clays are detailed in Table 1. Molten PDA was amended with clay–Az (1 mg L^−1^) for each treatment; clay without Az and unamended PDA served as the controls. For every treatment, five biological replicates (plates) were prepared. From the actively growing margin of 4–6-day-old *M. maydis* cultures, a 6 mm agar plug was excised and placed mycelium-side down at the plate center. Plates were incubated at 28 ± 1 °C in darkness for 6 days, and colony diameters were recorded in two perpendicular directions.

### 2.7. Mycelial Plug-Immersion Bioassay

A colony disk-immersion bioassay was employed to establish a calibration curve for Az. Commercial Amistar SC (25% *w*/*v* Az) solutions were prepared in sterile DDW at a range of concentrations and evaluated against representative *M. maydis* isolates at different immersion durations. Mycelial plugs (6 mm diameter) were excised from the actively growing margins of 4–6-day-old *M. maydis* cultures and immersed in the test solutions under sterile conditions for the designated time. The treated plugs were then placed centrally on 90 mm PDA plates (≥3 plates per concentration). Fungicide-free controls included both non-immersed mycelial plugs and plugs immersed in double-distilled water (DDW) for 10 s to assess any effect of the immersion procedure itself. Plates were incubated at 28 ± 1 °C in darkness for 3 and 6 days, after which colony diameters were measured along two perpendicular axes.

### 2.8. Slow-Release Kinetics of Azoxystrobin from Clay Carriers Using the Plug-Immersion Bioassay

To evaluate the slow-release behavior of Az from three clay carriers—bentonite (SW), sepiolite (S9), and MPO–Kaolinite (Table 2)—we prepared clay–Az formulations and subjected them to sequential aqueous washes. After each wash, two sample types were assayed: (i) the whole suspension (containing clay–Az particles dispersed in the liquid) and (ii) the clarified supernatant obtained by centrifugation at 3500 rpm for 10 min (representing Az released from the formulation). Parallel clay-only controls were included, tested either as particle suspensions or as their corresponding post-wash supernatants. Antifungal activity after each wash was quantified with the colony plug-immersion bioassay against *M. maydis*, and colony growth was recorded at 3 and 6 days of incubation at 28 ± 1 °C in darkness. Release performance was inferred by comparing bioactivity of the supernatants (released Az) to that of the corresponding particle-containing suspensions, and by benchmarking both against clay-only and Amistar SC solution (75 g L^−1^ Az) controls.

### 2.9. Statistical Analysis

Statistical analyses were conducted using GraphPad Prism software (version 10.6.0; GraphPad Software Inc., San Diego, CA, USA). Data normality was evaluated with the Shapiro–Wilk test, and most datasets satisfied the normality assumption (*p* > 0.05). Consequently, a one-way analysis of variance (ANOVA) was performed, followed by Fisher’s least significant difference (LSD) post hoc test, with statistical significance set at *p* < 0.05. For datasets that did not meet the normality assumption (*p* < 0.05), the nonparametric Kruskal–Wallis test was applied, followed by Dunn’s multiple comparisons test (uncorrected). For the slow-release kinetics of Az from clay carriers using the plug-immersion bioassay (Section 3.6), data were analyzed using a three-way analysis of variance (ANOVA) to evaluate the effects of treatment (clay type and formulation), wash number (1st vs. 2nd wash), and sampling day (Day 3 vs. Day 5), including all two-way and three-way interactions. When significant effects were detected, multiple comparisons among treatment means were performed using Tukey’s honestly significant difference (HSD) post hoc test.

## 3. Results

### 3.1. Azoxystrobin Growth Inhibition Assay for M. maydis Isolates

To evaluate the antifungal properties of the clay–azoxystrobin (Az) formulation, we first tested our *Magnaporthiopsis maydis* isolates in rich medium plates embedded with Az to identify variations in sensitivity and resistance to the fungicide (Figure 1). Indeed, after 3 days of incubation, the pathogen isolates revealed 38% variation in Az susceptibility, with Mm30, a highly aggressive strain, leading as the most resistant, and Mm25 (a moderate-high virulent strain) identified as the most susceptible. The sensitivity assays highlight the potential risk of resistance development.

### 3.2. Disk-Diffusion Assay

A follow-up assay was conducted to establish a calibration curve using the Az disk-diffusion method (Figure 2 and Figure 3). Although this assay is widely employed, it demonstrated limited sensitivity, as no significant growth inhibition was observed at Az concentrations below 3 mg L^−1^ (Figure 3; *p* > 0.05). Nevertheless, a clear dose-dependent response was detected within the concentration range of 4–10 mg L^−1^.

### 3.3. Clay–Az Formulations Mycelial-Growth Inhibition Assay

The growth inhibition assay (Figure 4) was used as an initial screening step to evaluate the efficacy of various clay–Az formulations prepared with different clay types (Table 1). All examined clays, either alone or combined with Az, significantly (*p* < 0.05) inhibited *M. maydis* colony growth by approximately 50% or more. In most cases, the addition of Az to the clay matrices significantly inhibited *M. maydis* growth compared with the corresponding clay-only treatments. However, bentonite–thiamine and sepiolite alone exhibited inhibitory activity comparable to that of their respective clay–Az combinations. Because direct particle–hypha contact in plate assays can contribute to growth suppression, this experiment was used primarily for qualitative screening. Conclusions regarding fungicide release were therefore based on the sequential wash plug-immersion assay, benchmarked against clay-only controls. Another exception was bentonite–berberine, which showed a weaker inhibitory effect in both forms, with or without Az. Based on these results, sepiolite and bentonite in their natural form were selected for subsequent stages of the study, including the development and optimization of the plug-immersion bioassay.

Because several matrices exhibited strong intrinsic antifungal activity, the incremental zone increase after Az addition was not always statistically significant under the plate conditions used. For subsequent mechanistic and release studies, we selected raw bentonite and raw sepiolite, which combined consistent bioactivity with simple, modifier-free preparation and reliable loading/release behavior.

### 3.4. Inhibition of M. maydis Spore Germination and Hyphal Growth by Clay–Az Formulations

The disk-diffusion assay was employed to evaluate the antifungal efficacy of the selected clay–Az formulations based on sepiolite and bentonite. In this assay, each formulation was placed at the center of a PDA plate, and the inhibition zone of *M. maydis* spore germination and hyphal growth was measured (Figure 5). The test was conducted under different conditions, with or without the addition of the antibiotic Hygromycin B (50 µg mL^−1^) to the medium and using either sterilized or non-sterilized clay preparations. The formulations remained stable after autoclaving. The results demonstrated that sterilization of the clays significantly enhanced their antifungal activity, while the presence of antibiotics had no adverse effect on their performance.

### 3.5. Mycelial Plug-Immersion Bioassay Calibration Curves

While the traditional inhibition assays described above provided useful preliminary information, their accuracy and sensitivity were limited. To overcome these limitations, a new mycelial plug-immersion bioassay was developed. In this method, *M. maydis* (isolate Mm2) mycelial plugs are immersed in test solutions (Figure 6A) and subsequently incubated on a rich solid medium at 28 °C for 3–6 days (Figure 6B). The initial calibration of this assay, performed with various Az concentrations (2–10 mg L^−1^) and immersion times (5–50 min), is shown in Figure 7A,B. Although the results demonstrated a clear dose- and time-dependent response—with higher concentrations and longer immersion times resulting in greater growth inhibition—they also indicated that the immersion period should be very short (on the order of seconds) and that the effective concentration range required further refinement. Therefore, the assay was repeated using immersion times of 1–120 s and Az concentrations ranging from 0.0001 to 1 mg L^−1^ (Figure 7C,D), which yielded a more sensitive and discriminating response.

To evaluate whether the immersion procedure itself affected fungal growth, non-immersed mycelial plugs were included as controls and compared with plugs immersed in DDW for 10 s. As shown in Table 3, colony growth of non-immersed controls was comparable to that of DDW-immersed plugs for both *M. maydis* isolates (Mm2 and Mm30) at both evaluation growth times (Days 3 and 6). No growth inhibition attributable to the immersion step was observed. Interestingly, isolate Mm30 showed a slight but significant increase in colony growth following DDW immersion (*p* < 0.05). These results indicate that the immersion step per se does not negatively affect mycelial growth and therefore does not confound the interpretation of treatment effects in the immersion assays.

Finally, calibration curves for the plug-immersion bioassay were established for two representative *M. maydis* isolates: the moderately aggressive isolate Mm2 and the highly virulent isolate Mm30. The immersion time was set to 10 s, and Az concentrations ranged from 0.01 to 10 mg L^−1^ (Figure 8). The data fitted the four-parameter logistic (4PL) model with excellent accuracy (R^2^ > 0.92), yielding the expected sigmoidal inhibition pattern with distinct upper and lower growth plateaus. The Day 3 curves (IC_50_ = 0.26 ± 0.03 mg L^−1^, slope = 2.4 for Mm2; IC_50_ = 0.30 ± 0.04 mg L^−1^, slope = 1.46 for Mm30) exhibited higher sensitivity compared with the Day 6 curves (IC_50_ = 1.50 ± 0.42 mg L^−1^, slope = 0.23 for Mm2; IC_50_ = 1.38 ± 0.40 mg L^−1^, slope = 1.37 for Mm30), indicating a time-dependent reduction in apparent fungicidal potency. This shift is consistent with the fungistatic nature of Az and the progressive colony expansion observed at sublethal concentrations during prolonged incubation. Accordingly, the Day 3 calibration curve, demonstrating greater sensitivity and a sharper inhibitory response, was selected as the reference function for quantifying effective Az concentrations released from the clay-based formulations in subsequent experiments.

### 3.6. Slow-Release Kinetics of Azoxystrobin from Clay Carriers Using the Plug-Immersion Bioassay

The slow-release behavior of Az from three clay carriers—bentonite, sepiolite, and MPO–Kaolinite—was assessed by subjecting prepared clay–Az formulations to two sequential aqueous washes and testing using the plug-immersion bioassay (Figure 9). A three-way ANOVA was performed to assess the effects of treatment (clay type and formulation), wash number (1st vs. 2nd wash), and sampling day (Day 3 vs. Day 5) on the measured response. Significant main effects were detected for treatment (*p* < 0.0001), wash number (*p* = 0.0208), and sampling day (*p* < 0.0001). Significant two-way interactions were observed between treatment × wash number (*p* < 0.0001) and treatment × sampling day (*p* < 0.0001), whereas the wash number × sampling day interaction was not significant (*p* = 0.1075). Importantly, a significant three-way interaction among treatment, wash number, and sampling day was detected (*p* < 0.0001), indicating that treatment-dependent responses differed between washes and sampling days. Overall, treatment accounted for the largest proportion of explained variance (54.8%), followed by sampling day (37.3%), whereas wash number contributed a smaller but significant fraction (0.13%).

After each wash, antifungal activity was evaluated separately in two sample fractions: (i) the complete suspension, containing clay–Az particles dispersed in the liquid phase, and (ii) the clarified supernatant obtained by centrifugation, representing the fraction of Az released into solution. All release-related interpretations are limited to qualitative and comparative bioactivity trends and do not represent direct chemical quantification of Az concentration. Compared to clay-only controls, which included each carrier type, tested both as particle suspensions and as their corresponding post-wash supernatants, the addition of Az resulted in a ca. 50% increase in antifungal activity. Based on the plug-immersion bioassay calibration (Figure 8), the antifungal activity observed for the released fractions was estimated to correspond to bioactivity equivalent to ≥1 mg L^−1^.

Comparison of bioactivity profiles between the particle-containing suspensions and the corresponding supernatants allowed qualitative assessment of relative release behavior among the different clay carriers. Specifically, in the bentonite and sepiolite formula, a similar bioactivity was detected in the supernatants and suspensions, indicating sustained bioactivity in both fractions, consistent with prolonged fungicide availability. This *M. maydis* suppression activity was similar to the inhibition that resulted when Az was added without the clay.

In contrast, in the MPO–Kaolinite treatment during the first wash, strong fungal inhibition was observed directly from the particle-containing suspension, whereas the supernatant exhibited a significant (*p* < 0.05) reduction in activity. This pattern suggests that only a minor fraction of the Az bound to the clay was released into the surrounding solution. However, in the second MPO–Kaolinite wash, pronounced antifungal activity was detected in both the particulate suspension and the supernatant, resulting in similarly high *M. maydis* inhibition levels to those observed for the bentonite and sepiolite formulations. Benchmarking these results against clay-only controls confirmed that the observed antifungal effects were attributable to the release of Az rather than intrinsic clay activity. Based on this step, bentonite and sepiolite were selected for the final evaluation.

Following the two-wash comparative assay (Figure 9), extended release profiling was conducted over seven sequential washes. Slow-release profiling of clay–Az formulations was conducted over a 144 h incubation period, with sequential washes performed every 24 h (seven washes in total, Figure 10). The results demonstrated a continuous release of Az, leading to strong inhibition of *M. maydis* (ca. 50%, *p* < 0.05) throughout the evaluation period. This inhibitory effect was comparable between the two fungal isolates tested (Mm2 and Mm30) and consistent across both bentonite- and sepiolite-based formulations. Moreover, the clay-only controls had no significant effect on fungal growth. When benchmarked against the calibration curve (Figure 8), the sustained antifungal activity indicated continued release of biologically active Az throughout the incubation period.

## 4. Discussion

Maize late wilt disease (LWD), induced by the phytopathogenic fungus *Magnaporthiopsis maydis*, exhibits endemic prevalence in Israel, Egypt, Spain, Portugal, India, and several other nations, yet remains relatively obscure on a global scale [4,5]. The ramifications of LWD in high-risk areas are profound, prompting the continuous pursuit of scientific endeavors to devise effective control strategies [19]. Presently, the primary management method is based on cultivating maize varieties exhibiting resistance to the pathogen. However, this approach faces challenges, as aggressive fungal strains can potentially overcome the immunity of these resistant cultivars [26,28]. Although azoxystrobin (Az) efficacy and the use of clays as generic carriers are individually well established, this study represents the first systematic investigation of clay-based carriers for the controlled release of Az targeting *M. maydis*. To our knowledge, no previous work has combined conventional antifungal assays with a dedicated, highly sensitive mycelial plug-immersion bioassay to characterize release kinetics and bioactivity of clay–fungicide formulations. By demonstrating that specific clays—particularly bentonite and sepiolite—can adsorb Az and release it gradually while maintaining strong antifungal efficacy, our results provide proof of concept for a new generation of sustainable fungicide delivery systems. These findings advance current knowledge beyond earlier studies on drip-irrigated Az applications [6,9,29,49] and lay the groundwork for environmentally friendly, field-oriented solution development that can complement or replace conventional chemical treatments.

Excessive pesticide use is a major source of environmental contamination, leaving behind large quantities of residues that either disperse into the environment or persist in crops until eventually reaching living organisms [50]. A key drawback of conventional formulations lies in their high content of inert components, such as surfactants and solvents, which may be even more toxic than the active substance itself [51,52]. This not only reduces overall efficacy but also heightens the risk of active ingredient accumulation in soil, air, and water, and ultimately in plants, animals, and humans. In recent decades, clay-based substrates have been widely studied as carriers to reduce the environmental footprint of pesticide use by minimizing leaching, degradation, and loss of efficacy [53,54,55,56,57,58,59].

The intrinsic antifungal activity of natural clays has been attributed in previous studies to several physicochemical factors, including ion exchange and surface interactions [60,61]. While these mechanisms were not examined directly here, clay-only controls confirmed that Az incorporation was required for sustained inhibition in the sequential wash and plug-immersion assays, despite some intrinsic antifungal activity observed in clay-only suspensions. Possible explanations reported in literature indicate a combination of physicochemical mechanisms, including: (i) ion-exchange, pH/redox buffering, and the release of metal cations (e.g., Fe^2+^, Al^3+^, Mg^2+^) that can be toxic to microbial cells; (ii) the high adsorption capacity of clays, which can locally reduce nutrient and water availability; and (iii) direct surface interactions between clay particles and microbial cells that may compromise cell-wall or membrane integrity. Similar inhibitory effects have been reported for sepiolite-based nanocomposites and bentonite soil amendments—the former suppressing rice-pathogen hyphae and spores, and the latter enhancing soil fungistasis in maize-field sandy soils [62,63]. Nevertheless, the incorporation of Az into these clay matrices markedly enhanced and prolonged antifungal activity, confirming that the dominant inhibition in our system results from sustained active-ingredient release rather than from the clays’ inherent properties. Moreover, clay–Az formulations showed sustained antifungal activity after repeated washes, demonstrating prolonged bioavailability compared with free Az. Extensive research has demonstrated the ability of diverse clay minerals to function as slow-release carriers for active compounds, thereby reducing leaching, evaporation, and photodegradation [64].

The enhanced antifungal activity observed for sterilized bentonite and sepiolite (Figure 5) does not result from thermal modification of their crystalline frameworks, which remain stable below 200 °C [65,66]. Instead, sterilization may remove residual organic matter or microbial biomass, indirectly improving formulation consistency and bioassay performance [67,68]. This cleaning effect improves dispersion in the growth medium and facilitates more uniform fungicide release. Consequently, the observed enhancement reflects physicochemical surface activation and microbial removal rather than mineralogical alteration.

Our initial screening revealed up to 39% variation in Az susceptibility among *M. maydis* isolates (Figure 1), reflecting genotypic diversity, as demonstrated before [25]. This variability raises concerns about the long-term effectiveness of chemical control and mirrors patterns observed in other filamentous fungi exposed to QoI fungicides [32,33]. Although sensitivity variation has not been documented to date for this pathogen, evidence from other soil-borne fungi indicates that repeated, high-dose application of single-site fungicides can select for resistant populations [69]. Azoxystrobin, like other strobilurins, disrupts fungal energy production by binding to the quinol oxidation (Qo) site of the cytochrome bc1 complex in mitochondria, thereby blocking electron transport and ATP synthesis [70]. However, resistance to QoI fungicides has emerged in over 20 fungal genera, including *Rhizoctonia solani*, *Alternaria alternata*, *Botrytis cinerea*, *Venturia inaequalis*, and *Mycosphaerella graminicola* [71]. Effective resistance management is therefore essential. Strategies include developing more potent fungicides, combining active ingredients with different modes of action, and integrating chemical and non-chemical approaches to reduce the risk of resistance in pathogens such as *M. maydis* [72,73]. Slow-release systems like those developed here could mitigate such risks by maintaining effective yet sublethal fungicide concentrations over extended periods, thereby reducing selection pressure on pathogen populations while ensuring continued disease suppression.

Traditional antifungal assays, including disk-diffusion and mycelial-growth inhibition tests, were valuable for initial screening but exhibited limited sensitivity—especially at sub-milligram Az concentrations—and failed to capture nuanced release dynamics. The pronounced activity of sepiolite in plate assays (Figure 4 and Figure 5) indicates that some carriers exhibit intrinsic fungistatic effects, which can mask incremental effects of Az loading under agar conditions. These limitations have been reported in other controlled-release studies, where conventional bioassays were found insufficient for accurately characterizing controlled-release or low-dose fungicidal effects [74,75], underscoring the need for more refined tools.

The plug-immersion bioassay developed here addresses this gap by offering superior sensitivity, a wider dynamic range, and the ability to resolve subtle dose–response relationships even with short exposure times. Such features are crucial when evaluating controlled-release formulations, where fungicide availability is often low but continuous—a scenario that closely mimics field soil environments where release occurs gradually around the rhizosphere. Despite these advantages, several methodological limitations should be acknowledged. It should be noted that the plug-immersion bioassay provides a relative, bioactivity-based proxy for Az availability rather than a direct measurement of fungicide uptake or adsorption, and short exposure times may introduce biological and operator-dependent variability.

The release behavior of Az (and other active ingredients) differs across clay carriers due to their structure and surface chemistry; smectitic clays (bentonite/montmorillonite) and fibrous sepiolite have well-documented adsorption–desorption features that govern loading and release kinetics. Multiple studies show sustained or controlled release from bentonite- and sepiolite-based formulations, with reduced leaching and prolonged availability in soils [41]. While the literature regarding Az binding and slow release from clays is scarce (see the review by [76]), conclusions can be drawn from other clay-pesticide studies. The release behavior of Az can vary markedly among different clay carriers, reflecting their distinct structural and physicochemical properties [77,78].

An example is kaolinite (MPO), which exhibited limited initial release (Figure 9), reflected by weak inhibition in the first supernatant wash, followed by stronger inhibition in the second wash. This wash-dependent pattern is consistent with delayed desorption or re-equilibration of bound Az rather than inconsistent behavior, and may relate to the non-swelling structure and comparatively low surface area of kaolinite, which can restrict early fungicide release. In contrast, bentonite and sepiolite exhibit a more constant and sustained release profile. Such controlled release is advantageous for field applications, as it can suppress early pathogen establishment in the rhizosphere and provide prolonged protection during the plant’s critical susceptibility period. In this context, the sequential-wash bioassay served as an in vitro proxy for fungicide leaching behavior, demonstrating the clay matrices’ ability to maintain localized, prolonged bioavailability and minimize immediate losses relative to soluble formulations. These findings emphasize that carrier mineralogy must be matched to the intended agronomic application, and that further modification (e.g., organo-functionalization [38,40]) could enhance retention–release balance and improve performance under field conditions.

The agronomic potential of clay-based Az carriers extends beyond their laboratory performance. Because clays are abundant, inexpensive, and environmentally benign [43], they can be readily integrated into existing farming practices—for example, as in-furrow strip placements, seed coatings, or soil amendments—without requiring new infrastructure. Importantly, such formulations could enhance current LWD control strategies that rely on resistant cultivars [20,21] or timed chemical applications [6,29], especially under conditions where these approaches are insufficient due to environmental variability or pathogen adaptation.

Moreover, sustained low-dose fungicide release could synergize with biological control agents such as *Trichoderma* spp. and beneficial rhizobacteria, improving their antagonistic activity against pathogens [6,30]. This is particularly relevant in real agricultural soils, where abiotic stresses and fluctuating moisture often reduce the efficacy of biocontrol agents. Controlled-release formulations thus provide a platform for true integrated disease management—one that combines chemical, biological, and cultural tools into a cohesive and environmentally sustainable strategy. To summarize, the bentonite- and sepiolite-based Az formulations developed in this study are expected, once validated in vivo, to represent a potential sustainable advancement over conventional Az use. By minimizing chemical inputs, reducing leaching and environmental residues, and enabling prolonged, localized disease suppression, these formulations align with integrated biological control strategies and promote a more environmentally responsible approach to crop protection.

## 5. Conclusions

In conclusion, this work provides the first demonstration that clay-based carriers can serve as controlled-release delivery systems for azoxystrobin (Az) against *M. maydis* under in vitro conditions. By combining high antifungal efficacy with controlled release, these formulations address key challenges associated with conventional fungicide use, including environmental contamination and rapid degradation, while highlighting their potential compatibility with integrated disease management strategies. The results provide a methodological basis for the development of next-generation crop protection technologies that may reduce chemical inputs and support the resilience of maize production systems in regions severely affected by late wilt disease. While the results presented here demonstrate controlled-release behavior in vitro, translating these findings to soil and field settings will require further investigation. Soil physicochemical properties, microbial activity, and plant root exudates may all influence Az adsorption and release dynamics, and these factors must be considered in future formulation designs. Field trials should assess the impact of slow-release systems on yield, disease progression, and pathogen population structure over multiple growing seasons. Furthermore, long-term monitoring of resistance evolution under controlled-release conditions will provide critical insights into the durability of this approach. Finally, exploring hybrid delivery systems—such as clay functionalized with biodegradable polymers or combined with nanomaterials—represents a potential future direction for enhancing release control and multi-pathogen efficacy.

## Figures and Tables

**Figure 1 jof-12-00021-f001:**
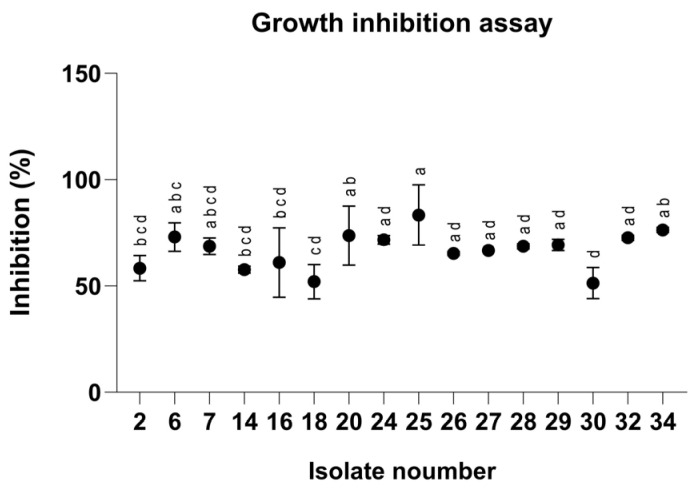
Inhibition percentages of *Magnaporthiopsis maydis* isolates grown on PDA medium (Potato Dextrose Agar, Difco, Detroit, MI, USA) supplemented with azoxystrobin at a concentration of 0.008 mg L^−1^ after 3 days of incubation. Values represent the mean ± standard error of three independent biological replicates. Different letters indicate significant differences (*p* < 0.05; one-way ANOVA with Fisher’s LSD). Values sharing at least one letter are not significantly different.

**Figure 2 jof-12-00021-f002:**
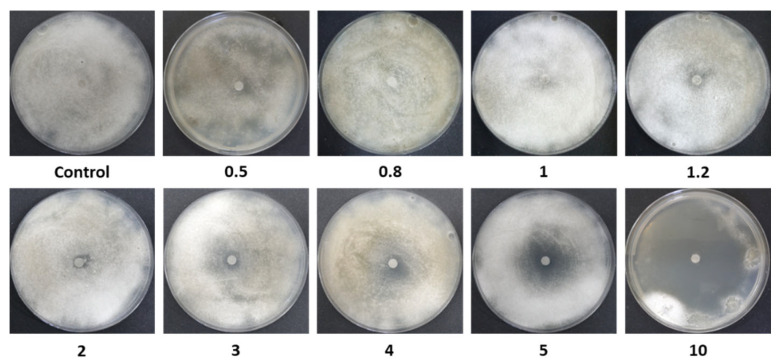
Disk-diffusion assay photos. Inhibition of spore germination and mycelial growth of *M. maydis* (isolate Mm2) on PDA, in the presence of azoxystrobin (Az) discs at different concentrations (0.5–10 mg L^−1^). Control plates contained discs without Az. Petri dishes were incubated in darkness at 28 ± 1 °C for 6 days. A representative image from three independent biological replicates is shown.

**Figure 3 jof-12-00021-f003:**
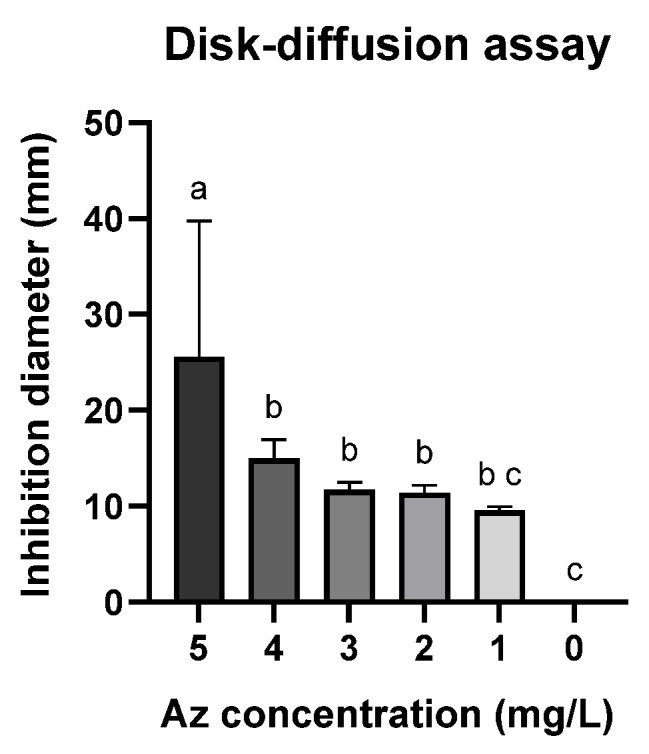
Disk-diffusion assay measurements showing the inhibition of spore germination and mycelial growth of *M. maydis* (Mm2 isolate) in the presence of azoxystrobin (Az) discs. Data represent the mean ± standard error of three independent biological replicates measured after 6 days of incubation at 28 ± 1 °C in darkness. The control (0) refers to a PDA disc without Az. Different letters (a–c) denote statistically significant differences among treatments (*p* < 0.05), determined using one-way ANOVA followed by Fisher’s least significant difference (LSD) post hoc test.

**Figure 4 jof-12-00021-f004:**
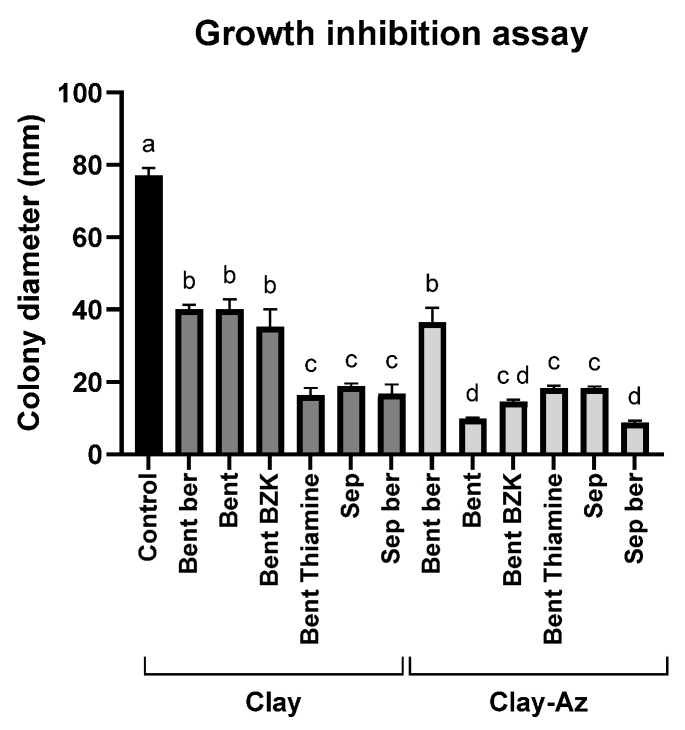
Growth inhibition assay of *M. maydis* (Mm2 isolate) on PDA medium supplemented with clay–azoxystrobin (Az) formulations. Mycelial discs were placed on PDA plates containing the formulations at a 1:2 pesticide-to-clay ratio. Data represent the mean ± standard error of five independent biological replicates measured after 6 days of incubation at 28 ± 1 °C in darkness. The control represents fungal growth on PDA medium supplemented with clay alone or without any addition. Different letters (a–d) indicate statistically significant differences among treatments (*p* < 0.05), as determined by one-way ANOVA followed by Fisher’s least significant difference (LSD) post hoc test.

**Figure 5 jof-12-00021-f005:**
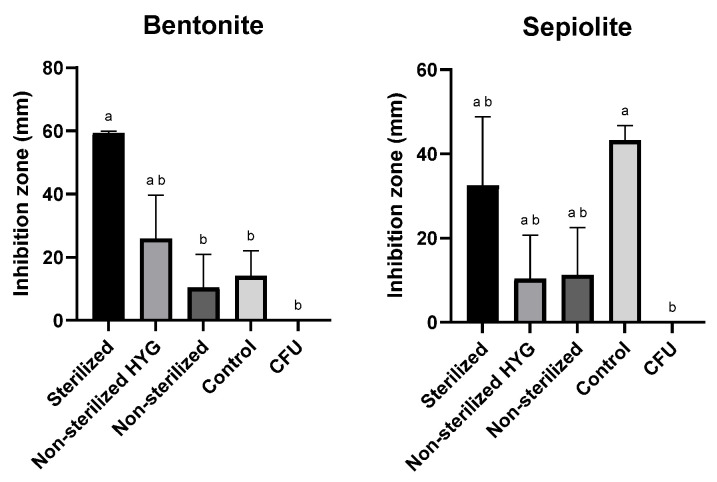
Inhibition of *Magnaporthiopsis maydis* (Mm2 isolate) spore germination and hyphal development in the presence of 50 mg clay–azoxystrobin (Az, commercial Amistar SC) formulation (prepared at a 1:5 (*w*/*w*) pesticide-to-clay ratio, equivalent to ~20% Az *w*/*w* in the dry formulation) or clay alone, placed at the center of PDA plates. Treatments were conducted with or without antibiotic supplementation (Hygromycin B, 50 µg mL^−1^) and using either sterilized or non-sterilized clay preparations. “CFU” denotes control plates without clay. Plates were incubated at 28 ± 1 °C in darkness for 6 days. Each treatment included three independent biological replicates. Data represent the mean ± SE. Different lowercase letters (a, b) above the bars indicate statistically significant differences (*p* < 0.05), determined using the Kruskal–Wallis non-parametric test followed by uncorrected Dunn’s multiple comparisons test.

**Figure 6 jof-12-00021-f006:**
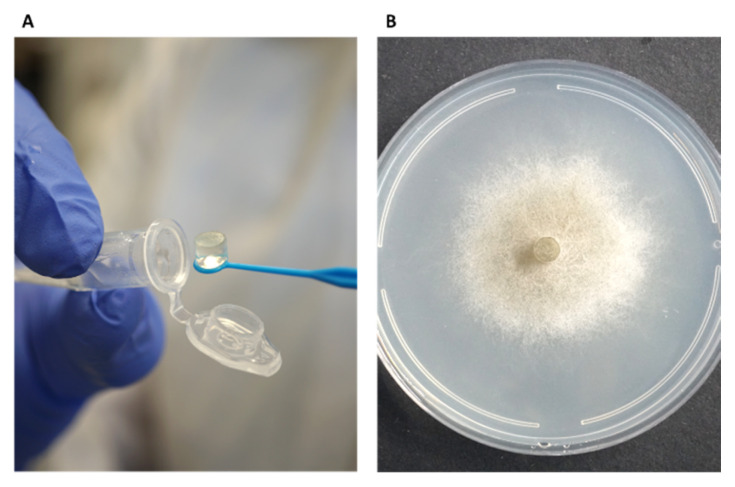
The mycelial plug-immersion bioassay was used to evaluate the antifungal activity of clay–azoxystrobin (Az) formulations against *M. maydis*. In this assay, mycelial plugs (6 mm diameter) excised from the actively growing margins of 4–6-day-old fungal cultures were immersed for 20 s in Az solutions (**A**) and then placed at the center of PDA plates. Plates were incubated in darkness at 28 ± 1 °C for 6 days, after which fungal growth inhibition was assessed (**B**). Mock control plates, seeded with non-immersed mycelial plugs or plugs immersed in double-distilled water (DDW), served as a reference to verify and standardize optimal growth conditions for *M. maydis*.

**Figure 7 jof-12-00021-f007:**
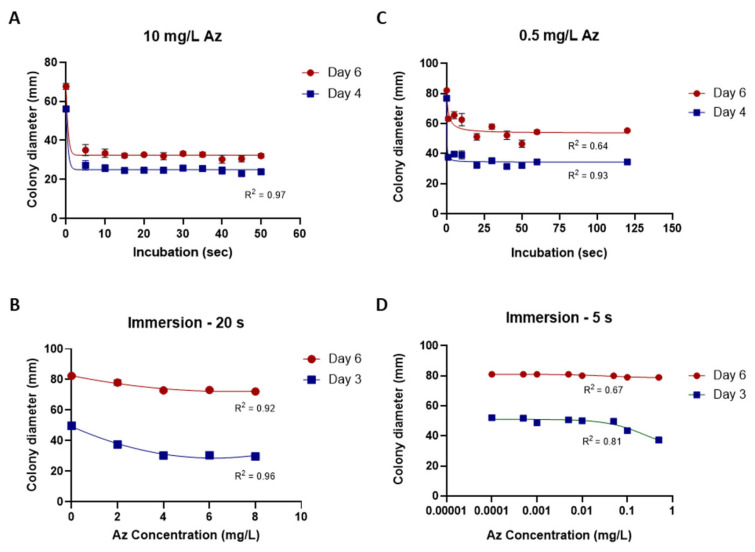
Calibration of the mycelial plug-immersion bioassay under different immersion times and azoxystrobin (Az) concentrations. (**A**) *Magnaporthiopsis maydis* (isolate Mm2) mycelial plugs were immersed in Az solution (10 mg L^−1^) for 5–50 s. (**B**) Plugs were immersed for 20 s in Az solutions containing 2–8 mg L^−1^. (**C**) Plugs were immersed in Az solution (0.5 mg L^−1^) for 1–120 s. (**D**) Plugs were immersed for 5 s in Az solutions containing 0.0001–1 mg L^−1^. Colony diameters were measured after 3 or 4, and 6 days of incubation at 28 ± 1 °C in darkness. Data represent the mean ± SE of 5 to 6 independent biological replicates. The antifungal activity data were fitted using a Padé (1,1) rational function. The resulting model exhibited a moderate to strong correlation with the experimental data, with coefficients of determination (R^2^) ranging from 0.64 to 0.97 across the various calibration curves.

**Figure 8 jof-12-00021-f008:**
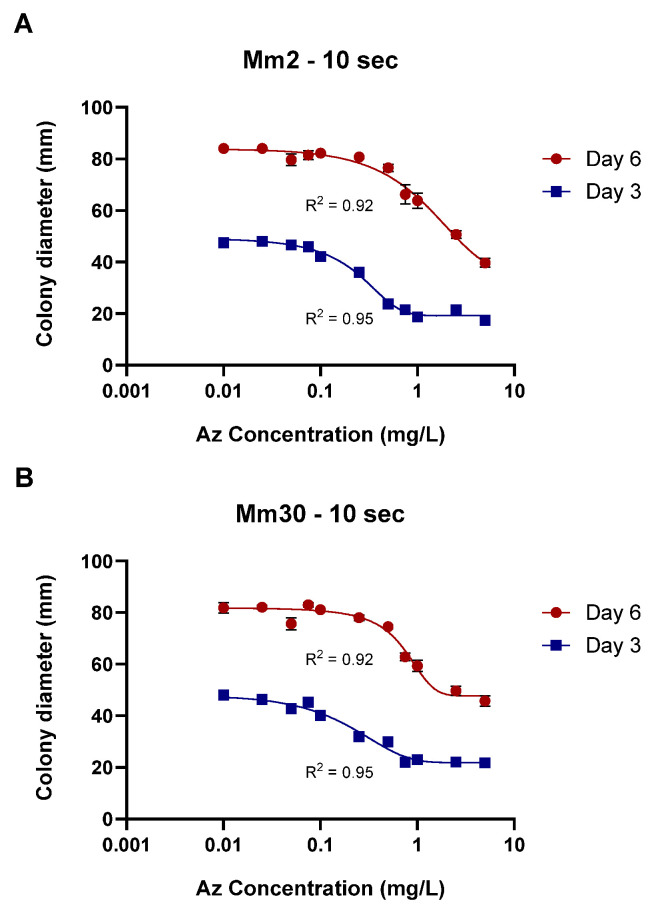
Calibration of the mycelial plug-immersion bioassay for two representative *Magnaporthiopsis maydis* isolates. Mycelial plugs of the moderately aggressive isolate Mm2 (**A**) and the highly virulent isolate Mm30 (**B**) were immersed in azoxystrobin (Az) solutions (0.01–10 mg L^−1^) for 10 s, and colony growth was recorded after 3 and 6 days of incubation at 28 ± 1 °C in darkness. Data represent the mean ± SE of five independent biological replicates. The antifungal activity data were fitted using a four-parameter logistic (4PL) model. The resulting model showed a strong correlation with the experimental data, with coefficients of determination (R^2^) ranging from 0.92 to 0.95 across the different calibration curves.

**Figure 9 jof-12-00021-f009:**
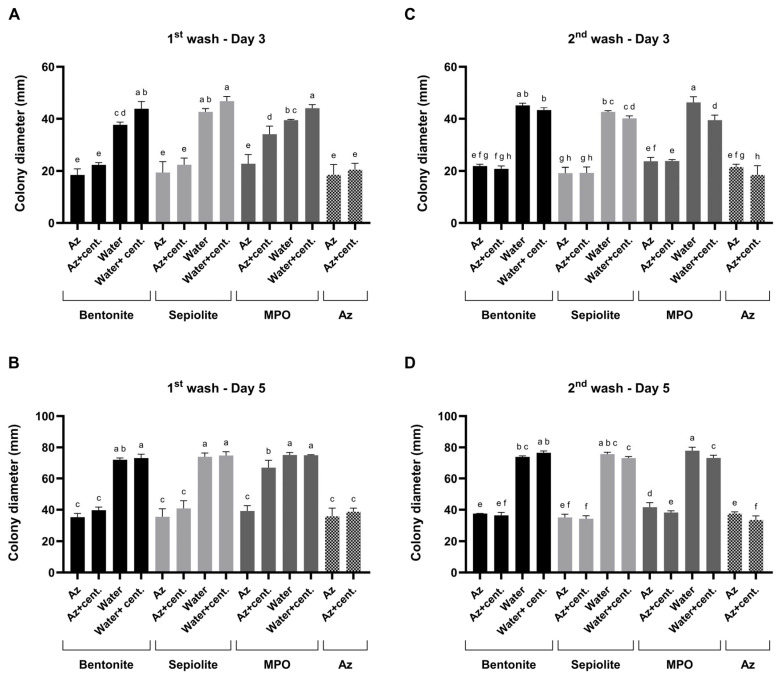
Mode of action profiling of clay–azoxystrobin (Az) formulations by two sequential washes. Bentonite, sepiolite, and kaolinite (MPO) clay–Az formulations were subjected to two consecutive aqueous washes to assess sustained fungicide release. (**A**,**B**)—Results of the first wash bioassay after 3 and 5 days of incubation. (**C**,**D**)—Results of the second wash bioassay after 3 and 5 days of incubation. After each wash, antifungal activity was evaluated separately for the particle-containing suspensions and their centrifuged (cent.) supernatants and compared with clay-only controls (both suspensions and post-wash supernatants) as well as with a diluted Amistar SC solution (75 g L^−1^ Az). Activity was determined using the colony plug-immersion bioassay against *M. maydis* (Mm2), where reduced colony growth indicates greater Az release or bioavailability. Data represent the mean ± SE of five independent biological replicates. Different letters indicate significant differences (*p* < 0.05; one-way ANOVA with Fisher’s LSD). Values sharing at least one letter are not significantly different.

**Figure 10 jof-12-00021-f010:**
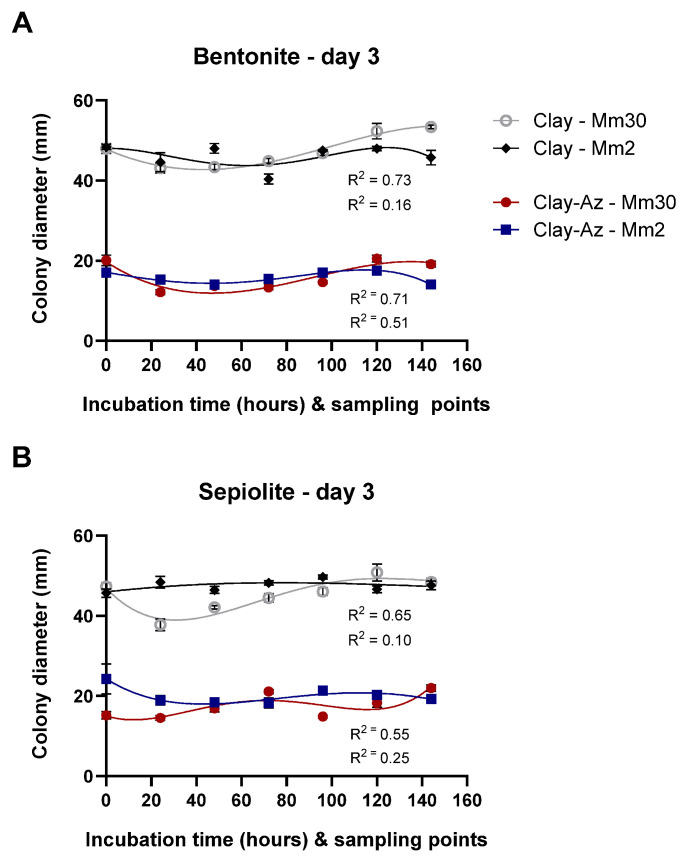
Slow-release profiling of clay–azoxystrobin (Az) formulations by sequential washes. Mycelial plugs of the moderately aggressive *Magnaporthiopsis maydis* isolate Mm2 and the highly virulent isolate Mm30 were immersed in bentonite–Az or bentonite alone (control) (**A**), and sepiolite–Az or sepiolite alone (**B**). The clay–Az formulations and clay-only controls were incubated at 28 ± 1 °C in darkness for 0–144 h. At selected time points (0, 24, 48, 72, 96, 120, and 144 h), the suspensions were centrifuged, and the supernatants were collected to evaluate antifungal activity. Following each wash, fresh double-distilled water was added to the clay–Az or untreated clay pellet, and incubation continued until the next sampling point. Fungal growth inhibition was assessed after 3 days of recovery on PDA under the same conditions. Data represent the mean ± SE of five independent biological replicates. A fourth-order polynomial regression (R^2^ = 0.10–0.76) showed a weak to moderate agreement between model predictions and the observed relationship between released Az concentration and fungal growth inhibition.

**Table 1 jof-12-00021-t001:** Examined Clays.

Clay Name	Type and General Information
**Sep–Sepiolite S9 (S9)**	Raw. Commercial sepiolite (Tolsa, Spain, CAS No: 63800–37-3), a magnesium–silicate mineral with a needle-like structure connected along the edges to form nanometer-sized channels. Exhibits excellent adsorption capacity, including uncharged molecules (primarily along the corners). It does not swell upon wetting and is chemically stable.
**Sep Ber–Sepiolite modified with Berberine (Sber)**	Organoclay. Sepiolite (S9) modified with berberine, a natural organic cation derived from Berberis shrubs. The berberine amount is adjusted to neutralize the negative charge of the raw clay, producing hydrophobic surfaces.
**Bent–Sigma Bentonite–Montmorillonite (SW)**	Raw. Commercial bentonite (Sigma–Aldrich, Rehovot, Israel, CAS No: 1302–78-9) is composed mainly of Na-montmorillonite (a 2:1 aluminosilicate mineral built of sheets). These sheets may swell when wetted or when large organic molecules are intercalated. The mineral adsorbs cations by an exchange mechanism, and in most cases does not adsorb considerable amounts of non-charged molecules.
**Bent BZK–Montmorillonite–Bentonite modified with Benzalkonium**	Organoclay. Bentonite (SW) modified with Benzalkonium (BZK), a quaternary ammonium cation known as a preservative that inhibits microbial development. The BZK content is adjusted to neutralize the negative charge of the raw clay, producing hydrophobic surfaces able to adsorb nonpolar molecules.
**Bent Thiamine–Montmorillonite–Bentonite modified with Thiamine B1 (BB1)**	Organoclay. Bentonite (SW) modified with Vitamin B1 (Thiamine), a cationic molecule containing an aromatic pyrimidine ring. The B1 content is adjusted to neutralize the negative charge of the raw clay, producing hydrophobic surfaces (as in Bent BZK).
**Bent Ber–Montmorillonite–Bentonite modified with Berberine (Bber)**	Organoclay. Bentonite (SW) modified with berberine (see Sber for details). The berberine content is adjusted to neutralize the negative charge of the raw clay, producing hydrophobic surfaces.
**MPO–Kaolinite**	Raw. Kaolinite (MPO—Minerals Processing Operations Ltd., Israel; CAS No. 1332-58-7), a natural aluminosilicate clay with a 1:1 layer structure, where sheets are held together by hydrogen bonds. It is non-swelling and widely used in the food and cosmetics industries.

**Table 2 jof-12-00021-t002:** Preparation of selected clay–Azoxystrobin Substrates.

Clay Source	Raw Dry Clay Weight (mg)	Clay–Az ^a^ Formulation Weight (mg)	Absorbed Solution Weight ^b^ (mg)	Absorbed Solution Volume (mL)	Absorbed Amistar SC Volume (mL)	Absorbed Amistar SC Weight (mg)	Amistar SC in the Formulation (%)	Az AI ^c^ Weight in Formulation (mg)	Az AI in Formulation (%)
**Bent (SW)**	20.2	94.3	74.1	0.072	0.036	38.3	40.7	9.0	9.6
**Sep (S9)**	19.7	92.1	72.4	0.070	0.035	37.5	40.7	8.8	9.6
**MPO–Kaolinite**	23.8	32.9	9.1	0.009	0.004	4.7	14.3	1.1	3.4

^a^ Az—azoxystrobin; ^b^ Density of pure Amistar SC (25% *w*/*v*, Az, Syngenta, Basel, Switzerland; supplied by Adama Makhteshim, Ashdod, Israel): 1062.8 mg mL^−1^, Amistar SC solution dilution: ×2, Az concentration in the solution: 125 mg mL^−1^; ^c^ AI = Active ingredient (Az).

**Table 3 jof-12-00021-t003:** Comparison of non-immersed and DDW-immersed mycelial plug controls on fungal growth. ^a^

Treatment	Day 3				Day 6			
	Mm2		Mm30		Mm2		Mm30	
	Mean	S.E.	Mean	S.E.	Mean	S.E.	Mean	S.E.
Control	49.9 ^C^	1	41.5 ^D^	1.9	83.5 ^A^	0.2	77.6 ^B^	2.4
DDW 10 s	53.1 ^C^	0.9	50.4 ^C^	0.7	83.5 ^A^	0.2	83.5 ^A^	0.2

^a^ The data represent the mean of 10 replications from two independent experiments. Different letters (A–D) indicate statistically significant differences among treatments (*p* < 0.05), as determined by one-way ANOVA followed by Fisher’s least significant difference (LSD) post hoc test.

## Data Availability

All the data generated or analyzed during this study are included in this published article.

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
