# Peer review of "J. Fungi2026, 12(1), 21;https://doi.org/10.3390/jof12010021"

_jof, 2025, doi:10.3390/jof12010021_

Round 1

Reviewer 1 Report

The manuscript presents a comprehensive study on clay-based controlled-release formulations of azoxystrobin for managing maize late wilt disease. The work is innovative, particularly in developing a plug-immersion bioassay capable of detecting low concentrations of fungicide and characterizing release kinetics from clay carriers. The topic is relevant to sustainable plant disease management and aligns well with the journal’s scope.

  1. The manuscript is excessively long and contains redundant background information, especially in the Introduction and Discussion, making it read partially like a review article rather than a focused research study. A 30–40% reduction in these sections is recommended.

  2. Claims regarding adsorption mechanisms, clay–fungicide interactions, and structural behavior are speculative because no physicochemical characterization (e.g., FTIR, XRD, SEM, BET, zeta potential) is presented. Mechanistic interpretations must be supported with data or significantly toned down.

  3. The experimental workflow lacks clear rationale. Multiple assays are used (disk-diffusion, growth inhibition, plug-immersion) without a clearly defined hypothesis-driven sequence. A clearer justification or experimental flow diagram would strengthen the study.

  4. The plug-immersion bioassay, although innovative, raises methodological concerns. Very short immersion times (1–10 seconds) introduce operator variability, uptake assumptions are unvalidated, and colony growth as an indirect proxy for fungicide uptake may be influenced by other biological factors. Limitations should be explicitly acknowledged.

  5. The release kinetics analysis is insufficient because it lacks direct chemical quantification. All conclusions about Az concentration are inferred from bioassay calibration curves, which compounds uncertainty. Polynomial fits with low R² values weaken interpretations. This section should emphasize qualitative patterns rather than quantitative estimates.

  6. Statistical analysis is simplistic for the complexity of the dataset. Multi-factor experiments (isolate × clay × wash × time) would be better supported using factorial ANOVA or mixed-effects models rather than repeated one-way ANOVA.

  7. Certain results are contradictory or insufficiently explained. Sepiolite shows strong intrinsic antifungal activity comparable to Az-loaded versions, complicating interpretation. Kaolinite shows weak initial release but sudden high inhibition in the second wash; this inconsistency requires clarification.

  8. Conclusions are overstated. Terms like “field-ready,” “proof of concept,” or “sustainable advancement” are premature without in-soil or greenhouse validation. Claims about integration with biological control are speculative unless demonstrated experimentally.

Author Response

Responses to Reviewer # 1’s comments

Thank you very much for all the comments that improved the quality of the manuscript. The time and effort invested are greatly appreciated. We have addressed them all and hope that the revised version meets your criteria for publication.

Major comments

The manuscript presents a comprehensive study on clay-based controlled-release formulations of azoxystrobin for managing maize late wilt disease. The work is innovative, particularly in developing a plug-immersion bioassay capable of detecting low concentrations of fungicide and characterizing release kinetics from clay carriers. The topic is relevant to sustainable plant disease management and aligns well with the journal’s scope.

Reply: Thank you for the positive and encouraging evaluation of our work.

Detailed comments

The manuscript is excessively long and contains redundant background information, especially in the Introduction and Discussion, making it read partially like a review article rather than a focused research study. A 30–40% reduction in these sections is recommended.

Reply: Thank you for this constructive comment. In response, we substantially streamlined the manuscript by removing major background and interpretative paragraphs from both the Introduction and Discussion. These sections were refocused to emphasize the study’s specific objectives, experimental rationale, and key findings, rather than providing broad review-style context. Details of the changes are outlined below and further addressed in our response to Comment 2.

Claims regarding adsorption mechanisms, clay–fungicide interactions, and structural behavior are speculative because no physicochemical characterization (e.g., FTIR, XRD, SEM, BET, zeta potential) is presented. Mechanistic interpretations must be supported with data or significantly toned down.

Reply: We agree with the reviewer that, in the absence of physicochemical characterization (FTIR, XRD, SEM, BET, zeta potential), mechanistic interpretations regarding adsorption and structural interactions are speculative. We have therefore revised the Abstract, Introduction, Results, and Discussion sections to remove or clearly qualify such statements, limiting interpretation to biologically demonstrated release behavior and framing mechanistic aspects as hypotheses for future study:

Abstract

To avoid implying structural and surface mechanisms, the sentence "A comparative analysis of particle suspensions and supernatants revealed formulation-specific release behaviors, likely linked to the mineral structure and surface properties", was replaced with: "A comparative analysis of particle suspensions and supernatants revealed formulation-specific release behaviors, which differed among clay carriers".  (lines 20-22). By that, we replace mechanistic attribution with phenomenological observation.

Introduction

We deleted a whole paragraph on clay mechanisms that includes general literature truths to avoid implicitly justifying a system mechanistically.

Instead, two sentences were combined in the text:

"Clay minerals have been widely reported to act as platforms for controlled pesticide delivery due to their diverse morphologies and sorption capacities [38-41]". (lines 76-77)

"In the present study, we did not directly characterize clay–fungicide interactions; instead, we evaluated their functional consequences through biological release assays". (lines 84-86)

Results

To avoid interpretation of binding and partitioning, we replaced the sentence: "suggesting that high effective Az was released into the test fluid while significant amounts of the fungicide remained associated with the clay matrix" with the following: "indicating sustained bioactivity in both fractions, consistent with prolonged fungicide availability". (lines 458-459)

Discussion

  1. The following sentence was replaced to frame as possible explanations reported in literature: "Natural clays showed measurable antifungal activity against maydis (Figures 4–5), consistent with previous reports demonstrating intrinsic antimicrobial effects of several clay minerals (e.g., [60,61]. The observed activity may arise from a combination of physicochemical mechanisms, including…". The sentence now reads: "The intrinsic antifungal activity of natural clays has been attributed in previous studies to several physicochemical factors, including ion exchange and surface interactions [60,61]. While these mechanisms were not examined directly here, clay-only controls confirmed that Az incorporation was required for sustained inhibition. possible explanations reported in literature indicate a combination of physicochemical mechanisms, including…". (lines 538-544)
  2. Since “activates” implies structural change, we replaced the sentence: "Instead, sterilization likely ‘activates’ the clays by removing adsorbed organics and resident microbes, increasing the availability of surface sites for azoxystrobin", with the following: "Instead, sterilization may remove residual organic matter or microbial biomass, indirectly improving formulation consistency and bioassay performance". (lines 561-563)
  3. The following FTIR-based mechanistic attribution paragraph was removed from the discussion: "The observed differences in azoxystrobin (AS) release kinetics between sepiolite and bentonite can be attributed to their distinct adsorption mechanisms, as supported by FTIR evidence (Rytwo G., personal communication). For sepiolite, the interaction appears to occur mainly at silanol (Si–OH) edge sites along the needles of the fibrous mineral. This is typically reflected in the alteration of the OH deformation doublet near 750–800 cm⁻¹ [79,80] and the modification of the Si–O–H stretching band at approximately 3700 cm⁻¹ [81]. These interactions, consistent with surface adsorption of neutral molecules, explain the strong binding and gradual desorption observed for the sepiolite–Az system. In contrast, adsorption of neutral organic molecules to bentonite (Na-montmorillonite) is less straightforward. Although AS lacks ionizable groups such as amides, its pyrimidine core and two phenoxy substituents enable van der Waals (π–π) stacking within the interlayer region and weak complexation with exchangeable cations on the clay surface. Such interactions have been documented for phenoxy herbicides [82] and planar pyrimidine structures [83,84]. The latter had shown a ~10 cm⁻¹ shift of the C–N ring stretching vibration (from ≈ 1560 cm⁻¹ to higher wavenumbers), consistent with cation-mediated complexation and interlayer stabilization of Az molecules."
  4. The following mechanistic attribution paragraph was removed from the discussion: "Taken together, these findings are consistent with previously described mechanisms, whereby sepiolite is reported to retain AS primarily through edge-site adsorption to silanol groups, while bentonite stabilizes the fungicide within its interlayer space through π-stacking and electrostatic interactions. These complementary mechanisms explain the sustained release and high antifungal efficacy of both clay formulations observed in the sequential-wash assays (Figures 9–10), strengthening the link between their physicochemical properties and biological performance.

The experimental workflow lacks a clear rationale. Multiple assays are used (disk-diffusion, growth inhibition, plug-immersion) without a clearly defined hypothesis-driven sequence. A clearer justification or experimental flow diagram would strengthen the study.

Reply: Thank you for this constructive comment. We agree that clarifying the experimental logic strengthens the manuscript. The study was designed as a stepwise, hypothesis-driven workflow in which each assay addressed a specific limitation of the previous one and informed the subsequent stage. We have now clarified this rationale explicitly in Section 2.1 (Rationale and research design) and improved the transitions between assays in the Results.

Briefly, the experimental sequence was as follows:

(i) Standard mycelial-growth inhibition assays were first used to characterize azoxystrobin (Az) sensitivity across a diverse set of M. maydis isolates and to identify representative strains for downstream analysis.

(ii) Disk-diffusion assays were then applied to establish a conventional dose–response framework and to demonstrate the limited sensitivity of classical agar-based methods at low Az concentrations—highlighting the need for a more sensitive approach suitable for controlled-release systems.

(iii) Clay–Az growth inhibition and disk-based assays served as an initial screening step to compare different clay carriers and exclude formulations with strong intrinsic antifungal activity or inconsistent behavior.

(iv) Mycelial plug-immersion bioassay was developed based on these limitations, to test the central hypothesis of the study: that clay carriers enable sustained, low-dose bioactive release of Az that cannot be accurately quantified using conventional plate assays. This assay was first calibrated using commercial Az solutions and then applied to quantify release kinetics from clay formulations via sequential-wash experiments.

To further improve clarity, we have emphasized that the plug-immersion bioassay is not a parallel assay but rather the analytical core of the study, specifically designed to resolve low-concentration release dynamics relevant to slow-release formulations. We agree that a schematic workflow would be useful and have therefore added a brief textual overview in the Methods; a graphical flow diagram can be included in a revised version if the editor recommends it.

A concise workflow explanation has been added to Section 2.1 to replace the current brief description.

"2.1. Rationale and research design

The experimental design followed a hypothesis-driven, stepwise workflow in which each assay addressed a specific question and informed the next stage. Conventional mycelial-growth inhibition and disk-diffusion assays were first used to (i) characterize azoxystrobin (Az) sensitivity among Magnaporthiopsis maydis isolates and (ii) evaluate the limitations of standard plate-based methods at low fungicide concentrations. These assays served as screening and benchmarking tools but proved insufficient to resolve low-dose or gradual fungicide release. Clay–Az formulations were then screened to select suitable carrier materials with consistent behavior and strong antifungal activity. Based on these findings, a mycelial plug-immersion bioassay was developed and calibrated to test the central hypothesis of the study: that clay carriers enable sustained, biologically effective release of Az at sub-milligram concentrations. This assay was subsequently applied to quantify release kinetics from selected clay carriers using sequential-wash experiments, providing a sensitive and mechanistically relevant assessment of controlled-release performance." (lines 117-131)

Also, a Graphical Abstract has been added to clarify the hypothesis-driven progression of assays and the rationale for developing the plug-immersion bioassay:

 Graphical abstract

Stepwise, hypothesis-driven workflow for evaluating clay-based controlled-release azoxystrobin formulations against Magnaporthiopsis maydis. Conventional plate assays were used for initial screening but lacked sensitivity at low fungicide concentrations, motivating the development of a plug-immersion bioassay to quantify sustained fungicide release from clay carriers.

The plug-immersion bioassay, although innovative, raises methodological concerns. Very short immersion times (1–10 seconds) introduce operator variability, uptake assumptions are unvalidated, and colony growth as an indirect proxy for fungicide uptake may be influenced by other biological factors. Limitations should be explicitly acknowledged.

Reply: We thank the reviewer for raising these important methodological considerations. We fully agree that the plug-immersion bioassay has inherent limitations, and these are now explicitly acknowledged in the revised manuscript. Specifically, we clarify that (i) very short immersion times (1–10 s) may introduce operator-dependent variability; (ii) fungicide uptake by the mycelial plug is inferred from downstream biological response rather than measured directly; and (iii) colony growth reflects an integrated outcome that may be influenced by fungal physiological state in addition to fungicide exposure.

To mitigate these constraints, all immersion steps were performed using a standardized protocol, fixed exposure times, and multiple biological replicates, and the assay was calibrated across a wide concentration range with high reproducibility (R² > 0.92). Importantly, we emphasize that the assay is intended as a comparative, bioactivity-based proxy for fungicide availability and sustained release, rather than a direct quantification of fungicide uptake or adsorption mechanisms. Corresponding text has been added to the Discussion sections to clearly define the scope, assumptions, and limitations of the assay:

"Despite these advantages, several methodological limitations should be acknowledged. It should be noted that the plug-immersion bioassay provides a relative, bioactivity-based proxy for Az availability rather than a direct measurement of fungicide uptake or adsorption, and short exposure times may introduce biological and operator-dependent variability." (lines 599-603)

The release kinetics analysis is insufficient because it lacks direct chemical quantification. All conclusions about Az concentration are inferred from bioassay calibration curves, which compounds uncertainty. Polynomial fits with low R² values weaken interpretations. This section should emphasize qualitative patterns rather than quantitative estimates.

Reply: We thank the reviewer for this important observation and fully agree that the release-kinetics analysis is based on biological inference rather than direct chemical quantification. Accordingly, we have revised this section to de-emphasize quantitative concentration estimates and instead focus on qualitative and comparative release patterns among the different clay carriers.

Specifically, we removed or softened statements implying precise azoxystrobin concentrations, clarified that all concentration values are approximate and assay-derived, and reframed the calibration curves as functional bioactivity benchmarks rather than chemical measurements. We also revised the discussion of polynomial fits, explicitly noting that lower R² values reflect biological variability and the limitations of indirect modeling, and that these fits are used solely to illustrate overall release trends rather than to support quantitative kinetic modeling.

  1. The following sentence was added to section 3.6: "All release-related interpretations are limited to qualitative and comparative bioactivity trends and do not represent direct chemical quantification of Az concentration". (lines 447-449)
  2. Original sentence: "According to the calibration curve (Figure 8), the amount of Az released from the clays during the assay was estimated to correspond to a concentration of at least 1 mg L⁻¹", was revised to: "Based on the plug-immersion bioassay calibration (Figure 8), the antifungal activity observed for the released fractions was estimated to correspond to bioactivity equivalent to ≥1 mg L⁻¹ Az." (lines 451-454), to emphasize that response measured is estimation based on comparative release behavior.
  3. Original sentence: "Comparison of bioactivity profiles between the particle-containing suspensions and the supernatants allowed inference of the release kinetics of Az from the different carriers", was revised to: "Comparison of bioactivity profiles between the particle-containing suspensions and the corresponding supernatants allowed qualitative assessment of relative release behavior among the different clay carriers." (lines 455-457)
  4. Original sentence: "Based on the calibration curve (Figure 8), the concentration of Az released during the assay was estimated to be equivalent to ≥1 mg L⁻¹", was revised to: "When benchmarked against the calibration curve (Figure 8), the sustained antifungal activity indicated continued release of biologically active Az throughout the incubation period". (lines 492-494)

Statistical analysis is simplistic for the complexity of the dataset. Multi-factor experiments (isolate × clay × wash × time) would be better supported using factorial ANOVA or mixed-effects models rather than repeated one-way ANOVA.

Reply: Thank you for this important comment. We agree that this approach explicitly tests main effects and interactions, and is more appropriate for the experimental structure. Therefore, we re-analyzed the dataset using a factorial (three-way) ANOVA in GraphPad Prism, with treatment (clay type/formulation), wash number (1st vs. 2nd wash), and sampling day (Day 3 vs. Day 5) as fixed factors, including all interaction terms, followed by Tukey’s HSD post hoc test (p < 0.05). 

We have updated the Methods–Statistics section and the Results to reflect the factorial analysis:

The following text was added to section 2.9. (Materials and Methods: Statistical analysis): "For the slow-release kinetics of Az from clay carriers using the plug-immersion bioassay (section 3.6), data were analyzed using a three-way analysis of variance (ANOVA) to evaluate the effects of treatment (clay type and formulation), wash number (1st vs. 2nd wash), and sampling day (Day 3 vs. Day 5), including all two-way and three-way interactions. When significant effects were detected, multiple comparisons among treatment means were performed using Tukey’s honestly significant difference (HSD) post hoc test." (lines 263-269)

The following text was added to section 3.6. (Results: Slow-release kinetics of azoxystrobin from clay carriers using the plug-immersion bioassay): "A three-way ANOVA was performed to assess the effects of treatment (clay type and formulation), wash number (1st vs. 2nd wash), and sampling day (Day 3 vs. Day 5) on the measured response. Significant main effects were detected for treatment (p < 0.0001), wash number (p = 0.0208), and sampling day (p < 0.0001). Significant two-way interactions were observed between treatment × wash number (p < 0.0001) and treatment × sampling day (p < 0.0001), whereas the wash number × sampling day interaction was not significant (p = 0.1075). Importantly, a significant three-way interaction among treatment, wash number, and sampling day was detected (p < 0.0001), indicating that treatment-dependent responses differed between washes and sampling days. Overall, treatment accounted for the largest proportion of explained variance (54.8%), followed by sampling day (37.3%), whereas wash number contributed a smaller but significant fraction (0.13%)." (lines 432-443)

Certain results are contradictory or insufficiently explained. Sepiolite shows strong intrinsic antifungal activity comparable to Az-loaded versions, complicating interpretation. Kaolinite shows weak initial release but sudden high inhibition in the second wash; this inconsistency requires clarification.

Reply: Thank you for highlighting these points. We agree that intrinsic clay bioactivity can complicate interpretation in plate-based screening assays. In our initial mycelial-growth inhibition screen (Figure 4), sepiolite alone showed inhibitory activity comparable to sepiolite-Az, which likely reflects strong particle–hypha contact effects under agar conditions and can reduce the apparent incremental effect of adding Az.

To avoid confusion, we now clarify in the Results/Discussion that this screening assay was used primarily to select candidate carriers, whereas conclusions on “release” are based on the sequential-wash plug-immersion bioassay in which each carrier was benchmarked against clay-only controls (both suspensions and post-wash supernatants (Figures 9, 10):

Results (lines 315-318): "Because direct particle–hypha contact in plate assays can contribute to growth suppression, this experiment was used primarily for qualitative screening. Conclusions regarding fungicide release were therefore based on the sequential wash plug-immersion assay, benchmarked against clay-only controls."

Discussion (lines 588-590): "The pronounced activity of sepiolite in plate assays indicates that some carriers exhibit intrinsic fungistatic effects, which can mask incremental effects of Az loading under agar conditions."

Regarding kaolinite (MPO), we agree that the wash-dependent pattern requires a clearer explanation. In wash 1, MPO showed strong inhibition in the particle-containing suspension but reduced activity in the supernatant, consistent with limited dissolved release and predominantly particle-bound bioactivity. In wash 2, strong inhibition was observed in both fractions, consistent with delayed desorption/re-equilibration leading to increased Az detected in the supernatant.

To address this, we replaced the short explanation in the Discussion: "An example is kaolinite (MPO), which showed limited initial release (Figure 9), likely due to its low surface area and non-swelling structure, which restricts fungicide desorption", with the following: "An example is kaolinite (MPO), which exhibited limited initial release (Figure 9), reflected by weak inhibition in the first supernatant wash, followed by stronger inhibition in the second wash. This wash-dependent pattern is consistent with delayed desorption or re-equilibration of bound azoxystrobin rather than inconsistent behavior, and may relate to the non-swelling structure and comparatively low surface area of kaolinite, which can restrict early fungicide release." (lines 613-618)

Conclusions are overstated. Terms like “field-ready,” “proof of concept,” or “sustainable advancement” are premature without in-soil or greenhouse validation. Claims about integration with biological control are speculative unless demonstrated experimentally.

Reply: We thank the reviewer for this important comment. We have revised the Discussion and Conclusions section to avoid premature or overstated claims regarding field readiness, sustainability, and integration with biological control. The conclusions are now explicitly framed in terms of in vitro controlled-release behavior, and references to field applicability, proof of concept, and integration with biological agents are presented as future research directions that require experimental validation. These changes ensure that the scope and implications of the study are accurately and conservatively represented.

Reviewer 2 Report

The topic of the manuscript seems interesting to me because it deals with helping to eradicate a fungal disease that causes significant losses in corn crops.

However, the novelty of the manuscript is unclear to me, since the efficacy of azoxystrobin is proven and clays are known to support compounds of interest. Could some data be provided on the efficacy of the clay-az preparation?

Furthermore, in the new mycelium immersion trial, is the mycelium immersed in the fungicide or in the fungicide supported on the clay? The aim is to evaluate the efficacy of the new clay-based preparation. Also, the graphs should include a control without immersion to show the effect of immersion on mycelium growth.

Please review the text, as there are typos on lines 56 and 191, as well as in Figure 7A, which should show seconds, not minutes. The DOIs could be included in the references.

Author Response

Responses to Reviewer # 2’s comments

We would like to sincerely thank you for your valuable and constructive feedback. The time and effort you invested in reviewing our work are greatly appreciated and have significantly contributed to improving the quality and clarity of the manuscript. All comments have been carefully addressed, and we hope that the revised version meets the journal’s standards and your expectations for publication.

Major comments

The topic of the manuscript seems interesting to me because it deals with helping to eradicate a fungal disease that causes significant losses in corn crops.

Reply: Thank you for the positive and encouraging evaluation of our work.

 However, the novelty of the manuscript is unclear to me, since the efficacy of azoxystrobin is proven and clays are known to support compounds of interest. Could some data be provided on the efficacy of the clay-Az preparation?

Reply: We agree that the antifungal activity of azoxystrobin and the general use of clays as carriers are well established. The novelty of this study lies in demonstrating that natural clays (bentonite and sepiolite) enable sustained, biologically effective release of azoxystrobin against M. maydis, a pathosystem for which clay-based delivery has not been previously evaluated. Importantly, the clay–Az formulations maintained ~50% growth suppression after repeated sequential washes and up to 144 h of incubation, whereas free azoxystrobin rapidly lost activity. Using a newly developed, highly sensitive mycelial plug-immersion bioassay, we show that the released fraction retained antifungal efficacy equivalent to ≥1 mg L⁻¹ azoxystrobin, confirming that the clay-Az preparations provide prolonged bioavailability rather than simple adsorption.

The following sentence in the Discussion was rewritten to clearly state that: "Although azoxystrobin (Az) efficacy and the use of clays as generic carriers are individually well established, this study represents the first systematic investigation of clay-based carriers for the controlled release of Az targeting M. maydis, the causal agent of LWD." (lines 516-519)

See also the Discussion part that already referred to your concern, including literature citations: "Multiple studies show sustained or controlled release from bentonite- and sepiolite-based formulations, with reduced leaching and prolonged availability in soils [41]. While the literature regarding Az binding and slow release from clays is scarce (see the review by [76]), conclusions can be drawn from other clay-pesticide studies. The release behavior of Az can vary markedly among different clay carriers, reflecting their distinct structural and physicochemical properties [77,78]." (lines 607-612)

Furthermore, in the new mycelium immersion trial, is the mycelium immersed in the fungicide or in the fungicide supported on the clay? The aim is to evaluate the efficacy of the new clay-based preparation.

Reply: Indeed, in the plug-immersion bioassay, mycelial plugs were immersed either in azoxystrobin solutions (for calibration) or in aqueous suspensions or supernatants of the clay–Az formulations to specifically evaluate fungicide release and bioavailability from the carrier.

This issue was already explained in Materials and Methods Section  2.7 (the mycelial plug-immersion bioassay calibration) and in Section 2.8: "Release performance was inferred by comparing bioactivity of the supernatants (released Az) to that of the corresponding particle-containing suspensions, and by benchmarking both against clay-only and Amistar SC solution (75 g L⁻¹ Az) controls." (lines 246-249)

The following sentence was added to the discussion to clarify the issue more: "Moreover, clay–Az formulations showed sustained antifungal activity after repeated washes, demonstrating prolonged bioavailability compared with free Az." (lines 554-556)

Also, the graphs should include a control without immersion to show the effect of immersion on mycelium growth.

Thank you for this important suggestion. We have now explicitly addressed this concern by adding a new Table 3. Non-immersed mycelial plugs were included as controls in all experiments to verify that the immersion step itself does not affect fungal growth. For clarity and simplicity, these data are presented in a separate table, while the original figures were left unchanged. As shown in Table 3, colony growth of non-immersed controls was comparable to that of plugs immersed in DDW for 10 s at both evaluation times (Days 3 and 6) and for both isolates (Mm2 and Mm30), indicating that immersion per se did not suppress mycelial growth. Notably, isolate Mm30 exhibited slightly enhanced growth following 10 s DDW immersion (p < 0.05). Together, these results confirm that the effects observed in the immersion assays reflect treatment-specific responses rather than an artifact of the immersion procedure itself.

Section 2.7 ( in the Materials and Methods was updated accordingly: "The treated plugs were then placed centrally on 90-mm PDA plates (≥3 plates per concentration). Fungicide-free controls included both non-immersed mycelial plugs and plugs immersed in double-distilled water (DDW) for 10 s to assess any effect of the immersion procedure itself. Plates were incubated at 28 ± 1 °C in darkness for 3 and 6 days, after which colony diameters were measured along two perpendicular axes." (lines 2301-235)

Also, the following explanation was added to the Result section:

"As shown in Table 3, colony growth of non-immersed controls was comparable to that of DDW-immersed plugs for both M. maydis isolates (Mm2 and Mm30) at both evaluation growth times (Days 3 and 6). No growth inhibition attributable to the immersion step was observed. Interestingly, isolate Mm30 showed a slight but significant increase in colony growth following DDW immersion (p < 0.05). These results indicate that the immersion step per se does not negatively affect mycelial growth and therefore does not confound the interpretation of treatment effects in the immersion assays." (lines 380-386)

Detailed comments

Please review the text, as there are typos on lines 56 and 191, as well as in Figure 7A, which should show seconds, not minutes. The DOIs could be included in the references.

Reply: We thank the reviewer for the careful reading and constructive comments. We thoroughly reviewed the entire manuscript and corrected all textual typos and inconsistencies. Specifically, we corrected typographical errors in lines 56 and 191, as well as in Figure 7A. Additionally, DOIs have been included in the reference list where available.